# Childhood immune imprinting to influenza A shapes birth year-specific risk during seasonal H1N1 and H3N2 epidemics

**Katelyn M. Gostic**[1¤]*, Rebecca Bridge[2], Shane Brady[2], Cécile Viboud[3], Michael Worobey[4], James O. Lloyd-Smith[1,3]*

**1** Dept. of Ecology and Evolutionary Biology, University of California, Los Angeles, Los Angeles, California, United States of America, **2** Arizona Department of Health Services, Phoenix, Arizona, United States of America, **3** Fogarty International Center, National Institutes of Health, Bethesda, Maryland, United States of America, **4** Dept. of Ecology and Evolutionary Biology, University of Arizona, Tucson, Arizona, United States of America

¤ Current address: Dept. of Ecology and Evolution, University of Chicago, Chicago, Illinois, United States of America
* kgostic@uchicago.edu (KMG); jlloydsmith@ucla.edu (JOL-S)

**Data Availability Statement:** All relevant data are available as Supporting Information files. All data are also archived alongside the full suite of code

## Abstract

Across decades of co-circulation in humans, influenza A subtypes H1N1 and H3N2 have caused seasonal epidemics characterized by different age distributions of cases and mortality. H3N2 causes the majority of severe, clinically attended cases in high-risk elderly cohorts, and the majority of overall deaths, whereas H1N1 causes fewer deaths overall, and cases shifted towards young and middle-aged adults. These contrasting age profiles may result from differences in childhood imprinting to H1N1 and H3N2 or from differences in evolutionary rate between subtypes. Here we analyze a large epidemiological surveillance dataset to test whether childhood immune imprinting shapes seasonal influenza epidemiology, and if so, whether it acts primarily via homosubtypic immune memory or via broader, heterosubtypic memory. We also test the impact of evolutionary differences between influenza subtypes on age distributions of cases. Likelihood-based model comparison shows that narrow, within-subtype imprinting shapes seasonal influenza risk alongside age-specific risk factors. The data do not support a strong effect of evolutionary rate, or of broadly protective imprinting that acts across subtypes. Our findings emphasize that childhood exposures can imprint a lifelong immunological bias toward particular influenza subtypes, and that these cohort-specific biases shape epidemic age distributions. As a consequence, newer and less "senior" antibody responses acquired later in life do not provide the same strength of protection as responses imprinted in childhood. Finally, we project that the relatively low mortality burden of H1N1 may increase in the coming decades, as cohorts that lack H1N1-specific imprinting eventually reach old age.

used to perform analyses and generate plots, at
https://zenodo.org/badge/latestdoi/160883450.

**Funding:** KMG was supported by the National
Institutes of Health (F31AI134017, T32-
GM008185). JOLS was supported by NSF grants
OCE-1335657 and DEB-1557022, SERDP RC-
2635, and DARPA PREEMPT D18AC00031. MW
was supported by the David and Lucile Packard
Foundation. The funders had no role in study
design, data collection and analysis, decision to
publish, or preparation of the manuscript.

**Competing interests:** The authors have declared
that no competing interests exist.

## Author summary

Influenza viruses of subtype H1N1 and H3N2 both cause seasonal epidemics in humans,
but with different age-specific impacts. H3N2 causes a greater proportion of cases in older
adults than H1N1, and more deaths overall. People tend to gain the strongest immune
memory of influenza viruses encountered in childhood, and so differences in H1N1 and
H3N2's age-specific impacts may reflect that individuals born in different eras of influenza
circulation have been imprinted with different immunological risk profiles. Another idea
is that H3N2 may be more able to infect immunologically experienced adults because it
evolves slightly faster than H1N1 and can more quickly escape immune memory. We ana-
lyzed a large epidemiological data set and found that birth year-specific differences in
childhood immune imprinting, not differences in evolutionary rate, explain differences in
H1N1 and H3N2's age-specific impacts. These results can help epidemiologists under-
stand how epidemic risk from specific influenza subtypes is distributed across the popula-
tion and predict how population risk may shift as differently imprinted birth years grow
older. Further, these results provide immunological clues to which facets of immune
memory become biased in childhood, and then later play a strong role in protection dur-
ing seasonal influenza epidemics.

## Introduction

Childhood influenza exposures leave an immunological imprint, which has reverberating, life-
long impacts on immune memory. Foundational work on original antigenic sin [1] and anti-
genic seniority [2] shows that individuals maintain the highest antibody titers against
influenza strains encountered in childhood. But how these serological patterns map to func-
tional immune protection, and shape birth year-specific risk during outbreaks, remains an
active area of inquiry. One open question is the breadth of cross-protection provided by
immune memory imprinted in childhood.

We define immune imprinting as a lifelong bias in immune memory of, and protection
against, the strains encountered in childhood. Such biases most likely become entrenched as
subsequent exposures back-boost existing memory responses, rather than stimulating de novo
responses [3]. By providing particularly robust protection against certain antigenic subtypes,
or clades, imprinting can provide immunological benefits, but perhaps at the cost of equally
strong protection against variants encountered later in life. For example, every modern influ-
enza pandemic has spared certain birth cohorts, presumably due to cross-protective memory
primed in childhood [4–10]. Recently, we showed that imprinting also protects against novel,
emerging avian influenza viruses of the same phylogenetic group of hemagglutinin (HA) as
the first childhood exposure [9,11]. Imprinting may additionally shape birth year-specific risk
from seasonal influenza [12–14], but the importance of broadly protective immunity in this
context is still being evaluated [15–17].

Until recently, narrow cross-protective immunity specific to variants of a single HA subtype
has been considered the primary mode of defense against seasonal influenza. Lymphocyte
memory of variable epitopes on the HA head (i.e. sites at which hemagglutinin antigens of dif-
ferent subtypes show limited homology) drives this narrow, within-subtype protection, which
is the main mechanism of protection from the inactivated influenza vaccine. But a growing
body of evidence shows protection may also be driven by memory of other influenza antigens
(e.g. neuraminidase, NA) [18–20], or by immune response to conserved epitopes, many of
which are found on the HA stalk [11,15,21–23]. Antibodies that target conserved HA epitopes

can provide broad protection across multiple HA subtypes in the same phylogenetic group [21,23,24], where HA group 1 contains hemagglutinin subtypes H1 and H2, while group 2 contains H3. Only three HA subtypes have circulated seasonally in humans since 1918, H1, H2 and H3; H1 and H2 belong to phylogenetic group 1, while H3 is in group 2 [11,22,25].

Current thinking stipulates that within a single host, imprinting probably induces multiple levels of bias in immune memory, to both conserved (broadly protective) and variable (narrowly protective) sites on various influenza antigens. The functional role of any single layer of imprinted immune memory depends on both immunodominance hierarchies and epidemic context. Here, we examine which layers of imprinted memory impact risk from seasonal influenza.

Different breadths of immunity are expected to act differently on influenza epidemiology. Within-subtype immunity to HA is known to shape seasonal influenza's epidemiology and evolution [26], but due to rapid decay in the face of antigenic drift, it would not be expected to shape cohort-specific imprinting protection across an entire human lifetime [27,28]. Conversely, broad, HA group-level immune memory arises when lymphocytes target conserved HA epitopes. Responses to these conserved epitopes should be more stable over time, and can play a strong role in defense against unfamiliar influenza strains (e.g. novel, avian or pandemic subtypes [11,21,23,24,29,30]), but are not traditionally though to act strongly against familiar, seasonal influenza subtypes. However, responses to the conserved HA stem have recently been identified as an independent correlate of protection against seasonal influenza [15], and might play a particularly strong role against drifted seasonal strains whose variable HA epitopes have become unrecognizable. Thus, childhood immune imprinting may determine which birth cohorts are primed for effective defense against seasonal strains with conserved HA epitopes characteristic of group 1 or group 2, or with variable HA epitopes characteristic of a particular subtype (H1, H2, etc.). A similar line of reasoning may apply to immunity against NA, although much less attention has been paid to this antigen.

Since 1977, two distinct subtypes of influenza A, H1N1 and H3N2, have circulated seasonally in humans, with striking but poorly understood differences in their age-specific impact [9,12–14,31]. These differences could be associated with childhood imprinting: older cohorts were almost certainly exposed to H1N1 in childhood (since it was the only subtype circulating in humans from 1918–1957), and now seem to be preferentially protected against modern seasonal H1N1 variants [9,12–14]. Likewise, younger adults have the highest probabilities of childhood imprinting to H3N2 (**Fig 1**), which is consistent with relatively low numbers of clinically attended H3N2 cases in these cohorts. Alternatively, differences in the evolutionary dynamics of H1N1 and H3N2 could explain the observed age profiles. Subtype H3N2 exhibits slightly faster drift in its antigenic phenotype than H1N1, and as a result, H3N2 may be better able to escape pre-existing immunity in immunologically experienced adults, whereas H1N1 may be relatively restricted to causing disease in immunologically inexperienced children [32].

We analyzed a large surveillance data set of relatively severe, clinically attended influenza cases to test whether cohort effects from childhood imprinting primarily act against variable epitopes, only providing narrow cross-protection against closely related HA or NA variants of the same subtype, or against more conserved epitopes, providing broad cross-protection across HA subtypes in the same phylogenetic group (**Fig 1A and 1B**). We fitted a suite of models to data using maximum likelihood and compared models using AIC. In a separate analysis, we considered the hypothesis that differences in evolutionary rate of H1N1 and H3N2, rather than imprinting effects, shape differences in age distribution. Our results have implications for long-term projections of seasonal influenza risk in elderly cohorts [13], who suffer the heaviest burdens of influenza-related morbidity and mortality, and whose imprinting status will shift through time as cohorts born during different inter-pandemic eras grow older.

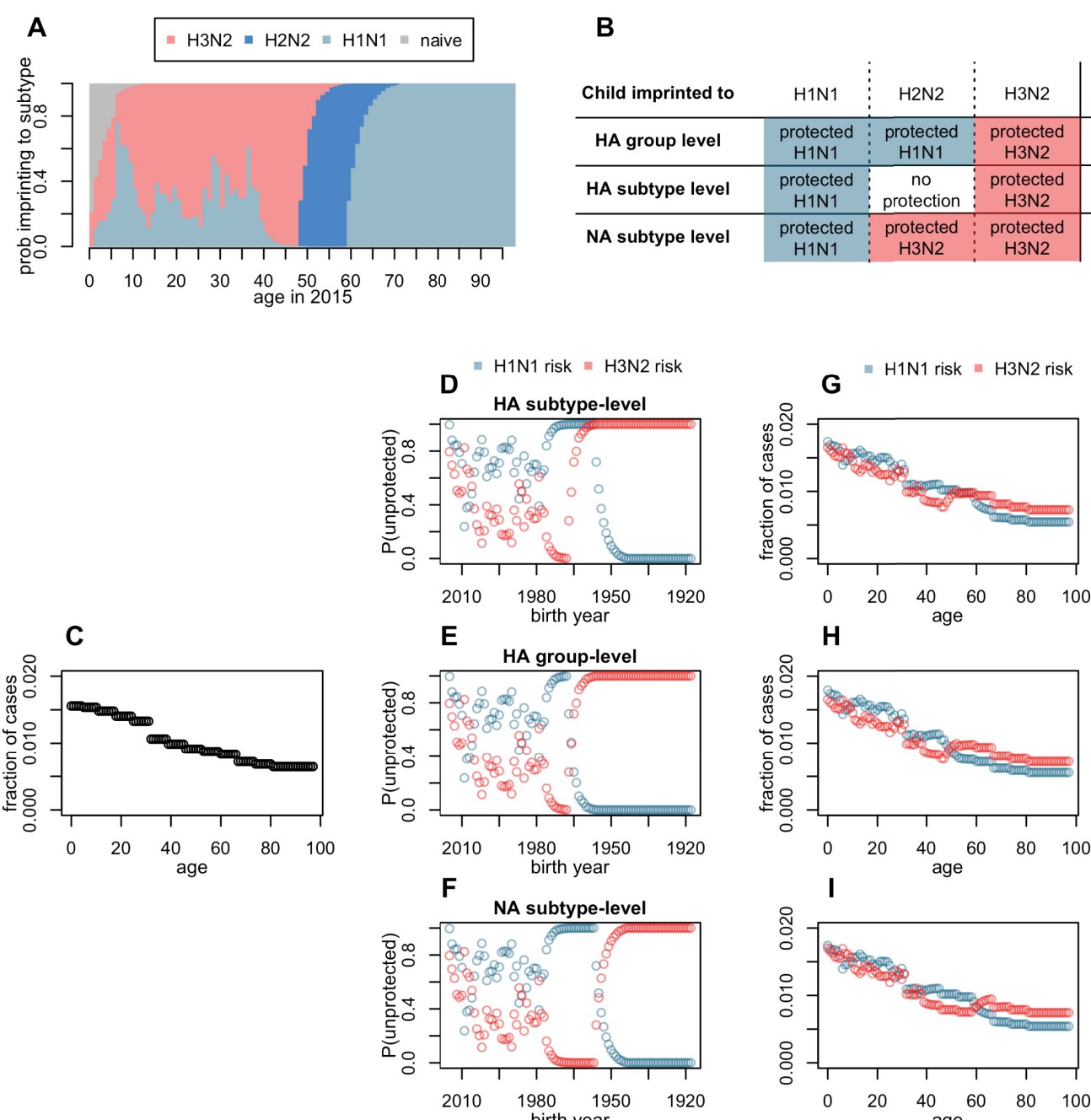

**Fig 1. Model and expectations under different imprinting hypotheses.** (**A**) Reconstructed, birth year-specific probabilities of imprinting (representative example specific to cases observed in 2015). Throughout the manuscript, group 1 HA subtypes are represented in blue and group 2 subtypes in red. (**B**) Expected imprinting protection against H1N1 or H3N2 under the three tested models. (**C**) Cartoon of expected age distribution of any influenza case, before controlling for subtype-specific imprinting. The shape of this curve is purely hypothetical, but each of our tested models combined demographic age distribution with a fitted, age-specific risk step function to generate similar, data-driven curves. (**D-F**) Fraction of each birth year unprotected by their childhood imprinting (from A) determines the shape of birth year-specific risk. (**G-I**) A linear combination of demography plus age-specific risk (as in C), and birth year-specific risk (as in D-F) give the expected age distribution of H1N1 or H3N2 cases under each model.

## Results

### Data

The Arizona Department of Health Services (ADHS) provided a dataset containing 9,510 seasonal H1N1 and H3N2 cases from their statewide surveillance system (**S1 Data**). Cases of all ages were confirmed to subtype by PCR and/or culture, primarily from virologic testing at the Arizona State Public Health Laboratory (ASPHL). The ADHS surveillance system aims to characterize circulating strains from patients across the state with medically attended influenza. Although surveillance does not target specific at-risk groups, relatively severe cases (especially those tested at hospital labs) are overrepresented in our data, as these cases are most likely to be medically attended and confirmed to subtype, thus meeting our study inclusion criteria. This is because confirmation to subtype requires a second-line test (PCR or culture); rapid tests are more common, but do not indicate subtype.

Although the exact collection setting of individual specimens was not always recorded, ADHS staff internally reviewed reporting source and provider organization of cases in our data to estimate that roughly 76% were reported and/or submitted by hospital labs or may have originated at hospital-associated outpatient clinics. In 2016, an ADHS analysis matched PCR-confirmed cases to hospital discharge data, and found that nearly half of the cases reported from hospital labs were severe enough to warrant hospital admission (if extrapolated to other seasons, roughly 38% of the overall data). The rest (also roughly 38% of the overall data, if extrapolated) were discharged without admission. To obtain a broader representation of clinically attended cases from across the state, ADHS collaborates with county health departments, commercial laboratories, and outpatient clinics to receive specimens. We estimate that roughly 8% of the overall data originated in outpatient settings. The remaining 17% of cases were either tested at commercial labs, or were tested at ASPHL, but with unknown origin. Ultimately, these data allow us to analyze drivers of relatively severe, clinically attended cases, but our results cannot be assumed to generalize to mild or asymptomatic cases.

Cases were observed across 22 years of influenza surveillance, from the 1993–1994 influenza season through the 2014–2015 season, although sample sizes increased dramatically after the 2009 pandemic (**Table 1**). Sampling changed slightly starting in 2004, when commercial labs

**Table 1. PCR-confirmed cases.**

| Season | Confirmed H1N1 | Confirmed H3N2 |
|---|---|---|
| 1993–94 | 0 | 101 |
| 1994–95 | 12 | 38 |
| 2002–03 | 71 | 8 |
| 2003–04 | 0 | 71 |
| 2004–05 | 0 | 131 |
| 2005–06 | 1 | 321 |
| 2006–07 | 212 | 28 |
| 2007–08 | 196 | 244 |
| 2010–11 | 472 | 1204 |
| 2011–12 | 595 | 348 |
| 2012–13 | 80 | 1578 |
| 2013–14 | 1475 | 151 |
| 2014–15 | 5 | 2109 |
| **Total** | **3119** | **6332** |

Data representing the first and second waves of the 2009 H1N1 pandemic (2008–2009 and 2009–2010 seasons) were excluded.

were first mandated to report positive tests to the state [33], but the vast majority of cases analyzed (9150/9451) were observed from the 2004–2005 season onwards, after this change had been implemented.

Following CDC standards, ADHS defines the influenza season as epidemiological week 40 (around early October) through week 39 of the following year [34]. The 2008–2009 and 2009–2010 influenza seasons spanned the first and second wave, respectively, of the 2009 H1N1 pandemic. We did not analyze cases observed during this time period, because age distributions of cases and immune memory differed during the 2009 pandemic from the normal drivers of seasonal influenza's immuno-epidemiology of interest to this study [14,21,24]. From the dataset of 9,510 seasonal cases (defined as any case observed outside the 2008–2009 or 2009–2010 season), we excluded 58 cases with birth years before 1918 (whose imprinting status could not be inferred unambiguously), and one case whose year of birth was recorded in error. Ultimately, we analyzed 9,541 cases.

## The model

**Reconstructed imprinting patterns.** We reconstructed birth year-specific probabilities of childhood imprinting to H1N1, H2N2 or H3N2 using methods described previously [11]. These probabilities are based on patterns of first childhood exposure to influenza A and reflect historical circulation (Fig 1A). Most individuals born between pandemics in 1918 and 1957 experienced a first influenza A virus (IAV) infection by H1N1, and middle-aged cohorts born between pandemics in 1957 and 1968 almost all were first infected by H2N2 (note that because the first influenza exposure may occur after the first year of life, individuals born in the years leading up to a pandemic have some probability of first infection by the new pandemic subtype, Fig 1A). Ever since its emergence in 1968, H3N2 has dominated seasonal circulation in humans, and caused the majority of first infections in younger cohorts. However, H1N1 has also caused some seasonal circulation since 1977, and has imprinted a fraction of all cohorts born since the mid-1970s (Fig 1A).

Reconstructions assumed children age 0–12 in the year of case observation might not yet have been exposed to any influenza virus. Interactions between imprinting and vaccination of naïve infants are plausible, but poorly understood [11,35]. We did not consider childhood vaccination effects here; only a small percentage of individuals in the ADHS data were born at a time when healthy infants were routinely vaccinated against influenza.

**Expected age distributions under alternate imprinting models.** If HA subtype-level imprinting protection shapes seasonal influenza risk, primary exposure to HA subtype H1 or H3 in childhood should provide lifelong protection against modern variants of the same HA subtype. If imprinting protection acts primarily against specific NA subtypes, lifelong protection will be specific to N1 or to N2 (Fig 1B). Alternatively, if broad HA group-level imprinting shapes seasonal influenza risk, then cohorts imprinted to HA subtype H1 or H2 (both group 1) should be protected against modern, seasonal H1N1 (also group 1), while only cohorts imprinted to H3 (group 2) would be protected against modern, seasonal H3N2 (also group 2) (Fig 1B). Collinearities between the predictions of different imprinting models (Fig 1D–1I) were inevitable, given the limited diversity of influenza antigenic subtypes circulating in humans over the past century (reflected in Fig 1A). Note that middle-aged cohorts, which were first infected by H2N2, are crucial, because they provide the only leverage to differentiate between imprinting at the HA subtype, NA subtype or HA group-level level (Fig 1B).

Our approach distinguishes between age-specific risk factors related to health and social behavior, and birth year-specific effects related to imprinting. Specifically, age-specific risk could be influenced by medical factors like age-specific vaccine coverage, age-specific risk of

severe disease, age-related changes in endocrinology and immunosenescence, or by behavioral factors like age-assorted social mixing, and age-specific healthcare seeking behavior. These factors should have similar impacts on any influenza subtype. In contrast, imprinting effects are subtype- (or group-) specific. Thus, we fit a step function to characterize the shape of age-specific risk of any confirmed influenza case. Simultaneously, we modeled residual, subtype-specific differences in risk as a function of birth year, to focus on the possible role of childhood imprinting. Each tested model used a linear combination of age-specific risk (**Fig 1C**) and birth year-specific risk (**Fig 1D–1F**) to generate an expected distribution of H1N1 or H3N2 cases (**Fig 1G–1I**). Note that for a given birth cohort, age-specific risk changed across progressive years of case observation (as the cohort got older), whereas birth year-specific risk was constant over time.

To test quantitatively whether observed subtype-specific differences in incidence were most consistent with imprinting at the HA subtype, NA subtype or HA group level, or with no contribution of imprinting, we fitted a suite of models to each data set using a multinomial likelihood and then performed model selection using AIC. AIC is used to compare the relative strength of statistical support for a set of candidate models, each fitted to the same data, and favors parsimonious models that fit the data well [36,37]. Technical details are provided in the **Methods**.

**Tested models.**   We fit a set of four models to the ADHS data set. The simplest model contained only age-specific risk, and more complex models added effects from imprinting at the HA subtype level, at the HA group level, or at the NA subtype level. The age-specific risk curve took the form of a step function, in which relative risk was fixed to 1 in age bin 0–4, and one free parameter was fit to represent relative risk in each of the following 12 age bins: {5–10, 11–17, 18–24, 25–31, 32–38, 39–45, 46–52, 53–59, 60–66, 67–73, 74–80, 81+}. Within models that contained imprinting effects, the fraction of individuals in each single year of birth with protective childhood imprinting was assumed proportional to reductions in risk. Two additional free parameters quantified the relative risk of a confirmed H1N1 or H3N2 case, given imprinting protection against that seasonal subtype.

**Effect of influenza evolutionary rate on age profiles.**   We used publicly available data from Nextstrain [38,39], and from one previously published study [40], to calculate annual antigenic advance, which we defined as the mean antigenic distance between strains of a given lineage (pre-2009 H1N1, post-2009 H1N1 or H3N2) that circulated in consecutive seasons (**Methods**). The "antigenic distance" between two influenza strains is used as a proxy for similarity in antigenic phenotype, and potential for immune cross-protection. A variety of methods have been developed to estimate antigenic distance using serological data, genetic data, or both [39–41].

To assess the impact of antigenic evolution on the epidemic age distribution, we tested whether the proportion of cases in children increased in seasons associated with large antigenic changes. If the rate of antigenic drift is a strong driver of age-specific influenza risk, then the fraction of influenza cases observed in children should be negatively related to annual antigenic advance [32]. In other words, strains that have not changed much antigenically since the previous season should be unable to escape pre-existing immunity in immunologically experienced adults, and more restricted to causing cases in immunologically inexperienced children; strains that have changed substantially will be less restricted to children.

## Subtype-specific differences in age distribution

Seasonal H3N2 epidemics consistently caused more clinically attended cases in older cohorts, while H1N1 caused a greater proportion of cases in young and middle-aged adults (**Figs 2, S1**

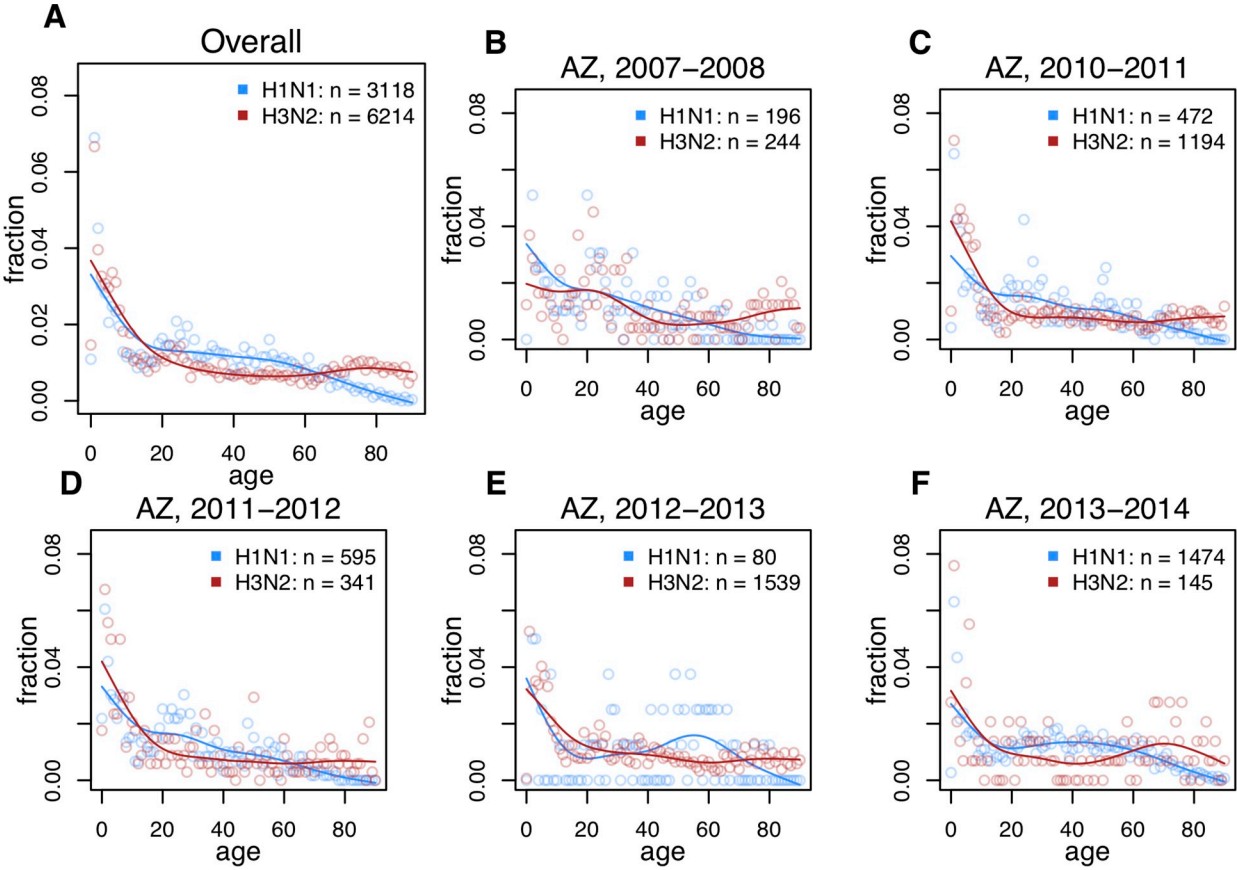

**Fig 2. Observed age distributions, Arizona.** Points show fraction of confirmed H1N1 or H3N2 cases observed in each single year of age. Lines show a smoothing spline fit to observed distributions. (**A**) All confirmed cases in the data (aggregate across all seasons). (**B-F**) Age distributions from individual seasons in which both H1N1 and H3N2 circulated (seasons with ≥ 50 confirmed cases of each subtype are shown here. See S1 Fig. for all seasons).

and **S2**). These patterns were apparent whether we compared H3N2 epidemic age distributions with those caused by the pre-2009 seasonal H1N1 lineage, or with the post-2009 lineage. Observed patterns are consistent with the predicted effects of cohort-specific imprinting (**Fig 1**), and with previously reported differences in age distribution of seasonal H1N1 and H3N2 incidence [12–14,31]. See **Fig 2** for seasons where H1N1 and H3N2 co-circulated in substantial numbers, and **S1** and **S2** **Figs.** for the entire dataset and alternate smoothing parameters.

## Imprinting model selection

The data showed a strong preference for NA subtype-level imprinting over HA subtype-level imprinting (ΔAIC = 34.54), and effectively no statistical support for broad, HA group-level imprinting (ΔAIC = 249.06), or for an absence of imprinting effects (ΔAIC = 385.42) (**Fig 3**, **Table 2**). Visual assessment of model fits (**Fig 3C and 3D**) confirmed that models containing imprinting effects at the narrow, NA or HA subtype levels provided the best fits to data. The lack of support for the no-imprinting model suggests that imprinting is important, and shapes lifelong seasonal influenza risk, just as it does avian-origin influenza (10, 12). However, imprinting appears to act more narrowly against seasonal influenza than against avian influenza, providing cross protection only to a specific NA or HA subtype, instead of broader, HA group-level protection. This result is consistent with the idea that immunodominance of

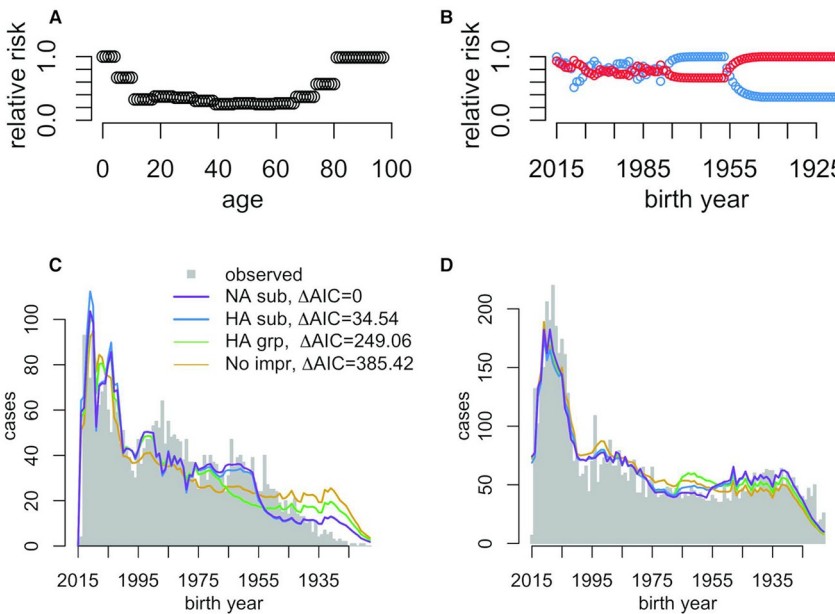

**Fig 3. Model fits and model selection. (A)** Fitted effects of age, after controlling for demographic age distribution and **(B)** imprinting effects from the NA subtype-level imprinting model, which provided the best fit to data. **(C-D)** Model fits to observed age distributions of H1N1 **(C)** and H3N2 **(D)** case. All models included demographic age distribution and age-specific risk.

variable HA epitopes limits the breadth of immune cross protection deployed against familiar, seasonal influenza subtypes [23,24].

**Table 2. Maximum likelihood parameter estimates and 95% profile confidence intervals.**

| Model | NA subtype-level imprinting | HA subtype-level imprinting | HA group-level imprinting | No imprinting |
|---|---|---|---|---|
| ΔAIC | 0.00 | 34.54 | 249.06 | 385.42 |
| H1N1 impr. protection | 0.36 (0.30–0.44) | 0.29 (0.24–0.35) | 0.65 (0.56–0.76) | |
| H3N2 impr. protection | 0.66 (0.58–0.76) | 0.90 (0.78–1.04) | 0.70 (0.62–0.82) | |
| Ages 0–4 | Reference group: Value fixed to 1 | | | |
| Ages 5–10 | 0.67 (0.62–0.73) | 0.65 (0.60–0.70) | 0.65 (0.60–0.71) | 0.61 (0.56–0.66) |
| Ages 11–17 | 0.33 (0.30–0.37) | 0.30 (0.28–0.34) | 0.32 (0.30–0.36) | 0.29 (0.27–0.33) |
| Ages 18–24 | 0.37 (0.34–0.42) | 0.34 (0.32–0.38) | 0.37 (0.34–0.42) | 0.34 (0.31–0.38) |
| Ages 25–31 | 0.35 (0.32–0.40) | 0.33 (0.30–0.38) | 0.34 (0.32–0.38) | 0.32 (0.29–0.36) |
| Ages 32–38 | 0.3 (0.28–0.35) | 0.28 (0.26–0.32) | 0.3 (0.27–0.34) | 0.27 (0.25–0.31) |
| Ages 39–45 | 0.25 (0.22–0.30) | 0.22 (0.20–0.26) | 0.25 (0.22–0.29) | 0.23 (0.21–0.26) |
| Ages 46–52 | 0.27 (0.24–0.30) | 0.22 (0.20–0.26) | 0.26 (0.23–0.29) | 0.24 (0.22–0.28) |
| Ages 53–59 | 0.25 (0.23–0.30) | 0.22 (0.20–0.26) | 0.23 (0.21–0.27) | 0.23 (0.20–0.26) |
| Ages 60–66 | 0.27 (0.24–0.30) | 0.29 (0.26–0.33) | 0.24 (0.22–0.28) | 0.23 (0.21–0.27) |
| Ages 67–73 | 0.37 (0.33–0.43) | 0.42 (0.37–0.48) | 0.34 (0.30–0.38) | 0.33 (0.30–0.38) |
| Ages 74–80 | 0.57 (0.50–0.64) | 0.64 (0.57–0.74) | 0.52 (0.46–0.59) | 0.5 (0.46–0.57) |
| Ages 81+ | 0.99 (0.88–1.11) | 1.12 (1.00–1.26) | 0.9 (0.81–1.01) | 0.87 (0.80–0.96) |

All estimated parameters represent the relative risk of a confirmed case, given the factors listed in the left-hand column. Age-specific risk parameters could take any positive value. Imprinting parameters could take values in [0,1], consistent with reductions in risk from the imprinted subtype or group. All tested models included age-specific risk and demographic age distribution.

As expected (see **Fig 1G–1I**), predictions from the two best models were highly collinear, except in their risk predictions among middle-aged, H2N2-imprinted cohorts (birth years 1957–1968), and some other minor differences arising from normalization across birth-years.

## Fitted risk patterns

Fitted age-specific risk curves took similar forms in all tested models. After controlling for demographic age distribution, estimated age-specific risk was highest in children and the elderly, consistent with the buildup of immune memory across childhood, and waning immune function in the aged (**Fig 3A** shows the fitted curve from the best model). Estimates of imprinting parameters were less than one, indicating some reduction in relative risk (**Table 2**). Within the best model, estimated reductions in relative risk from childhood imprinting were stronger for H1N1 (0.34, 95% CI 0.29–0.42) than for H3N2 (0.71, 95% CI 0.62–0.82). In the second-best model, HA subtype-specific imprinting, estimated reductions in H3N2 risk were particularly weak, and the confidence interval overlapped the null value of 1. **Table 2** shows parameter estimates and 95% profile confidence intervals from all models fitted.

## Effect of evolutionary rate

To test for effects of evolutionary rate on epidemic age distribution, we searched for decreases in the proportion of cases among children in seasons associated with antigenic novelty, when highly drifted strains might be more able to infect immunologically experienced adults. We defined children as ages 0–10, and verified internally that our analysis of evolutionary rate was insensitive to our exact choice of age range for children. Consistent with this expectation, the data showed a slight negative but not significant association between annual antigenic advance and the fraction of H3N2 cases observed in children (**Fig 4A**). However, note that no clear relationship emerged between antigenic novelty and the fraction of cases observed in older children (>10) and adults (**Fig 4A**). These are the cohorts in which epidemiological data show the clearest differences between H1N1 and H3N2's age-specific impacts (**Fig 2**); if rate of antigenic evolution is a dominant driver of age-specific differences in incidence, we would have expected to see clearer evidence of evolutionary rate effects among adults cohorts, not just between adults and the youngest children. The data contained too few influenza seasons with sufficient numbers of confirmed H1N1 cases to support meaningful Spearman correlation coefficients for either pre-2009 or post-2009 seasonal H1N1 lineages.

Furthermore, if evolutionary rate is the dominant driver of subtype-specific differences in epidemic age distribution, then when subtypes H1N1 and H3N2 show similar degrees of annual antigenic advance, their age distributions of cases should appear more similar. However, the data showed that differences in H1N1 and H3N2's age-specific impacts did not converge when lineages showed similar annual advance. When comparing the fraction of cases observed in specific age classes, H1N1 data consistently clustered separately from H3N2, with H1N1 consistently causing fewer cases at the extremes of age (children 0–10 and elderly adults 71–85), but more cases in middle-aged adults, regardless of antigenic novelty (**Fig 4A**). Smoothed density plots showed no clear relationship between annual antigenic advance and age distribution (**Fig 4B**). Overall, the data showed a weak, but not significant signal that relatively severe, clinically attended cases may be more restricted to young children when antigenic novelty is low, but the data did not show strong evidence that the magnitude of annual antigenic drift is a systematic driver of epidemic age distribution across the entire population.

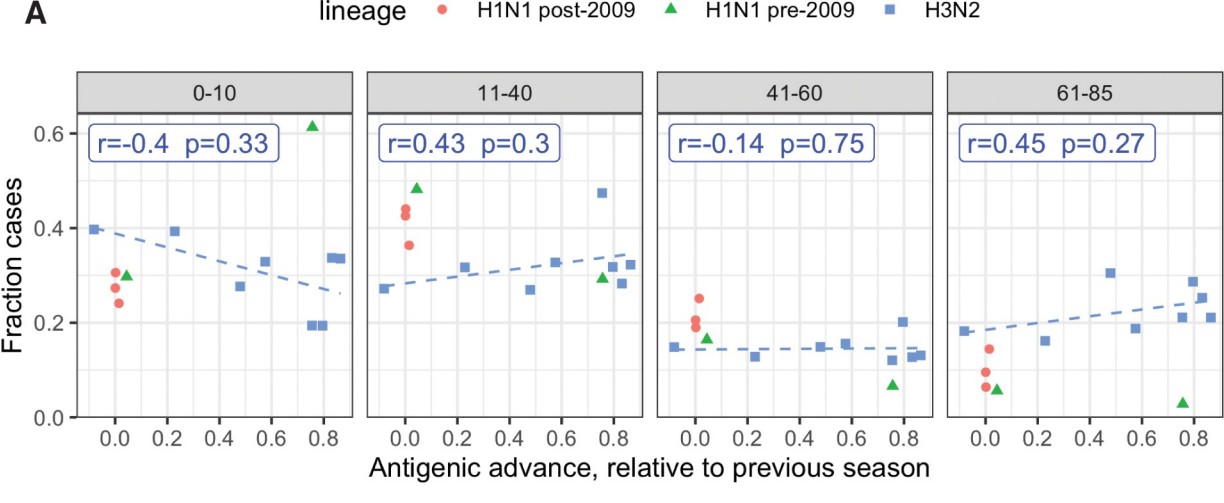

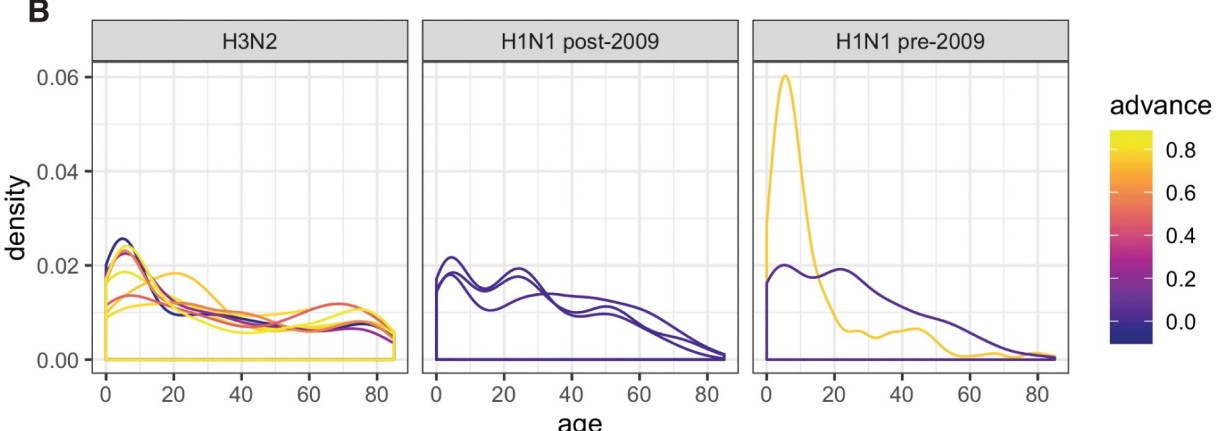

**Fig 4. Effect of antigenic advance on age distribution. (A)** Relationship between annual antigenic advance and the fraction of cases observed in children (0–10), or in adult age groups. Each data point represents a single influenza season in which at least 100 confirmed cases of a given subtype were observed. Blue label shows Spearman correlation between the fraction of H3N2 cases observed in each age group and annual antigenic advance. Blue dashes show linear trend fitted using lm() in R. **(B)** Season-specific age distributions of cases, colored by antigenic advance since the previous season.

## Discussion

We analyzed a large epidemiological surveillance dataset and found that seasonal influenza subtypes H1N1 and H3N2 cause different age distributions of relatively severe, clinically attended cases, confirming previously reported patterns [12–14]. We analyzed several possible drivers of these differences systematically, and found the greatest support for imprinting protection against seasonal influenza viruses of the same NA or HA subtype as the first influenza strain encountered in childhood [12,13]. The data did not support strong effects from broader HA group-level imprinting, as recently detected for novel zoonotic or pandemic viruses [9,11], or from differences in rates of antigenic evolution [32]. Our results suggest individuals retain a lifelong bias in immune memory, and that this imprint is not erased even after decades of exposure to or vaccination against dissimilar influenza subtypes.

External evidence corroborates the idea that birth year, rather than age, drives subtype-specific differences in seasonal influenza risk. When H3N2 first emerged in 1968, it caused little or no excess mortality in the elderly, who had putatively been exposed, as children or young

adults, to an H3 virus that had circulated in the late 1800s [7,9]. Meanwhile, H1N1-imprinted cohorts (those ~10–50 years old at the time) experienced considerable excess mortality in the 1968 pandemic [7]. Now, fifty years later, the same H1N1-imprinted cohorts continue to experience excess H3N2 morbidity and mortality as older adults [12–14,31] (**Fig 2**).

In model comparison, the data supported childhood imprinting to NA at the subtype level. Although NA is not as intensively studied as HA, these results emphasize the increasingly recognized importance of both antigens as drivers of protection against seasonal influenza [18–20]. Realistically, some combination of effects from both HA and NA subtype-level imprinting probably shapes seasonal influenza risk; both models of imprinting produced similar fits to data, and far outperformed other models in terms of AIC (Fig 3). Unfortunately, due to the limited diversity of seasonal influenza subtypes that have circulated in humans over the past century, collinearities between even the relatively simple models tested here prevented us from testing more complicated models of combined effects from imprinting to multiple antigens. Deeper insights into the respective roles of HA and NA will most likely need to come from focused immunological cohort studies, in which individual histories of influenza infection are recorded and can be studied alongside changes in serology, PBMCs, and/or the B cell repertoire [35]. Alternatively, the development of immunological biomarkers for diagnosis of imprinting status in individual patients could substantially increase the power of epidemiological inference.

We did not detect a clear relationship between annual antigenic advance and epidemic age distribution, although small sample sizes may have limited our statistical power. We did detect a weak trend, consistent with the idea that influenza cases are more restricted to immunologically inexperienced children in seasons of low antigenic advance, as previously proposed [32]. But the data did not reveal a clear relationship between antigenic advance and the fraction of cases occurring in adult age groups, where epidemiological data reveal distinct subtype-specific differences in impact. Perhaps antigenic advance shapes how cases are distributed between children and adults, but has small or inconsistent impacts within the adult population. We speculate that clearer relationships between antigenic advance and epidemic age distribution might emerge if methods to estimate antigenic distance were able to incorporate effects such as immune history [42], glycosylation [42,43], and immunity to antigens other than HA [19,20,44].

The exact immunological drivers of imprinting protection against seasonal influenza remain unclear, but our results provide some new clues. Traditionally, within-subtype cross-protection is thought to decay quickly with antigenic drift. Strains that circulated more than 14 years apart rarely show measurable cross-protective titers by the hemagglutination inhibition (HI) assay [40]. The short timescale of immune memory to variable HA head epitopes stands in contrast to patterns observed in our study and others [12–14], where within-subtype immune memory imprinted in childhood appears to persist for an entire human lifetime, remaining evident even in the oldest cohorts. We speculate that within-subtype imprinting protection may involve epitopes that are more conserved, and stable over time, than those typically measured in HI assays. These inform most existing estimates of antigenic distance, but disproportionately measure antibodies to variable, immunodominant epitopes on the HA head [15,24]. Across a lifetime of exposures to diverse H1N1 and H3N2 variants, repeated back-boosting of antibodies to intermediately conserved sites on HA or NA (i.e. sites conserved within but not across HA and NA subtypes), could explain the longevity of subtype-level imprinting protection. This is consistent with recent evidence that the immune repertoire shifts to focus on more conserved influenza epitopes as we age [27,28].

Another possibility is that memory B cell clones developed during the first childhood influenza infection may later adapt via somatic hypermutation to follow antigenic targets as they

drift over time. However, this would be inconsistent with new evidence suggesting memory B cells are relatively fixed in phenotype, and have little potential for ongoing affinity maturation [45,46], or that somatic hypermutation decreases with age [27]. Finally, the role of CD4+ T cells in imprinting is unclear, but T cell memory and T cell help to B cells within germinal centers both play at least some role in the development of the immune repertoire [47].

Signals of imprinting protection are anomalously strong in the current cohort of elderly adults, as reflected by higher estimates of imprinting protection to H1N1 than H3N2. The oldest subjects in our data, born slightly after 1918, and would not have encountered an influenza virus of any subtype other than H1N1 until roughly age 30. Repeated early-life exposures to diverse H1N1 variants may have reinforced and expanded the breadth of H1N1-specific immune memory [5,48]. But this strong H1N1 protection seems to come at a cost; even after decades of seasonal H3N2 exposure, and vaccination, older cohorts have evidently failed to develop equally strong protection against H3N2. HA group 1 antigens (e.g. H1) appear to induce narrower immune responses, and less cross-group protection than structurally distinct HA group 2 antigens (e.g. H3) [25]. Perhaps elderly cohorts imprinted to group 1 antigens have been trapped in narrower responses that offer exceptional protection against strains similar to that of first exposure but relatively poor adaptability to other subtypes.

We speculate that imprinting protection, which currently limits the number of severe, clinically-attended H1N1 cases in the elderly, also limits the mortality impact of H1N1 viruses. Although pre- and post- 2009 H1N1 lineages have caused slightly different profiles of age-specific mortality [13], neither H1N1 lineage causes nearly as many deaths as H3N2 in high-risk elderly cohorts [13,31,49]. On the one hand, if strong subtype-specific biases from imprinting remain in future cohorts of elderly adults, our results would corroborate the idea that mortality from H1N1 may increase as protection in the elderly shifts from H1N1 toward other subtypes [9,13]. On the other hand, given that cohorts born after 1968 have had much more varied early life exposures to both H1N1 and H3N2, these cohorts may show a greater ability to act as immunological generalists as they become elderly, capable of effective defense against multiple subtypes.

Our study has several limitations. Relatively severe, clinically attended cases are much more likely to be detected, confirmed to subtype, and included in our data than mild cases. Thus, while our results show a clear relationship between subtype-level imprinting and risk of relatively severe, clinically attended influenza, the relationship between imprinting and mild or asymptomatic cases could not be determined from available data.

Given the limited number of variables recorded in the data, we could not model explicitly the impact of individual risk factors such as the presence of comorbidities, patient sex, or vaccination status. All these factors are known to shape immunity and influenza risk [50], and all may cause individual imprinting outcomes to vary from the average, population trends measured by our study. Understanding how these patient-level covariates modulate imprinting and other aspects of immunity is the next frontier in this line of research. For now, working within the constraints of the available data, we designed the age-specific risk component of the model to capture empirically the combined effects of several risk factors that could not be modeled individually. Additionally, we analyzed the relative count of H1N1 to H3N2 cases within each single year of birth, not absolute incidence, to control for minor age-specific biases in sampling, which are almost inevitably present in any large surveillance data set.

Another limitation was the low number of confirmed cases available in the pre-2009 era. Large, detailed data sets collected continuously over decades provide the greatest power to separate the effects of age from birth year. We emphatically echo earlier calls [51] for more systematic sharing of single year-of-age influenza surveillance data, standardization of sampling effort, and reporting of age-specific denominators, which could substantially boost the

scientific community's ability to link influenza's genetic and antigenic properties with epidemiological outcomes. Additionally, collection and reporting of covariates such as sex, vaccination status and the presence of comorbidities in surveillance data would help us understand how patient-level variables modulate imprinting, and immunity in general [52,53].

Altogether, this analysis confirms that the epidemiological burden of H1N1 and H3N2 is shaped by cohort-specific differences in childhood imprinting [9,12,13,16,54], and that this imprinting acts at the HA or NA subtype level against seasonal influenza [16,17]. The lack of support for broader, HA group-level imprinting effects emphasizes the consequences of immunodominance of influenza's most variable epitopes, and the difficulty of deploying broadly protective memory B cell responses against familiar, seasonal strains. Overall, these findings advance our understanding of how antigenic seniority shapes cohort-specific risk during epidemics. The fact that elderly cohorts show relatively weak immune protection against H3N2, even after living through decades of seasonal exposure to or vaccination against H3N2, suggests that antibody responses acquired in adulthood do not provide the same strength or durability of immune protection as responses primed in childhood. Immunological experiments that consider multiple viral exposures, and cohort studies in which individual histories of influenza infection are tracked from birth, promise to illuminate how B cell and T cell memory develop across a series of early life exposures. In particular, these studies may provide clearer insights than epidemiological data into which influenza antigens, epitopes and immune effectors play the greatest role in immune imprinting, and how quickly subtype-specific biases become entrenched across the first or the first few exposures.

## Materials and methods

### Ethics statement

This study analyzed only existing epidemiological data, which was completely anonymized.

### Estimation of age from birth year in ADHS data

The data contained three variables, influenza season, birth year and confirmed subtype. For most cases, birth year was extracted directly from the reported date of birth in patient medical records, but age was not known. We estimated patient age at the time case observation using the formula [year of observation]-[birth year]. To ensure that the minimum estimated age was 0, the second year in the influenza season of case observation was considered the calendar year of observation (e.g. 2013 for the 2012–2013 season).

### Splines

In **Fig 2**, smoothing splines were fit to aid visual interpretation of noisy data. We fit splines using the command smooth.spline(x = AGE, y = FRACTIONS, spar = 0.8) in R version 3.5.0. Variables AGE and FRACTIONS were vectors whose entries represented single years of age, and the fraction of cases observed in the corresponding age group. The smoothing parameter 0.8 was chosen to provide a visually smooth fit. Alternative smoothing parameter choices (0.6 & 1.0) are shown in **S1 and S2 Figs**. Although the choice of smoothing parameter changed the shape of each fitted spline, qualitative differences between splines fitted to H1N1 or H3N2 were insensitive.

### Model formulation

For each unique season in which cases were observed, define p as a vector whose entries represent the expected probability that a randomly drawn H1N1 or a randomly drawn H3N2 case

was observed in an individual born in year b. Each model defined p as a linear combination of age-specific risk, birth year-specific risk (i.e. imprinting effects) and demographic age distribution. All tested models were nested within the equation:

$$p = DA * \mathbf{1}_{H1N1}(I_{H1N1}) * \mathbf{1}_{H3N2}(I_{H3N2}) \qquad 1$$

To include risk factors that only modulated risk from one subtype, we included indicator functions $\mathbf{1}_{H1N1}$ and $\mathbf{1}_{H3N2}$, which took value 1 if p described the expected age distribution of H1N1 or H3N2 cases, respectively, and 0 otherwise.

**Demographic age distribution (D).** The population of Arizona aged slightly across the study period, so we controlled for shifting demography in all tested models. Demographic age distribution was obtained from intercensal estimates of total population (both sexes) for the state of Arizona, based on the 2000 and 2010 census [55]. The US Census Bureau reports population estimates for ages 0–84, but only provides an aggregate estimate for ages 85+. We impute the number of individuals in each single year of age over 85 using a linear model fit to data on age 75–84, with a minimum threshold of 1000 individuals per single year of age. State-specific population estimates were not available prior to the 2000 census, so we substituted estimates from the year 2000 for cases observed in the 1993–94, and the 1994–95 seasons. Vector D represented the fraction of the total population at the time of case observation that fell in a given birth year.

**Age-specific risk (A).** Age-specific risk was defined as a step function, in which relative risk was fixed to value 1 in an arbitrarily chosen age bin, and then z-1 free parameters, denoted $r_2$ to $r_z$, were fit to describe relative risk in all other age bins. Below, $\mathbf{1}_i$ are indicator functions specifying whether each vector entry is a member of age bin i.

$$A = \mathbf{1}_1 + \mathbf{1}_2 r_2 + \cdots \mathbf{1}_z r_z \qquad 2$$

To obtain the predicted fraction of cases observed in each single year of birth, we normalized so that the product of vectors representing demographic age distribution, and age-specific risk, (DA in Eq 1) summed to 1. Thus, vector DA can be interpreted as the expected distribution of cases of any influenza case (either subtype), in the absence of birth year-specific biases from imprinting.

**Imprinting (I).** An indicator function defined whether a given prediction vector described risk of confirmed H1N1 or H3N2. Let $f_{IHxNy}$ be vectors describing the fraction of cases of each birth year that were protected against strain HxNy by their childhood imprinting. We defined $r_{IHxNy}$ as free parameters describing the risk of confirmed HxNy, given imprinting protection. Finally, the factor describing the effect of imprinting (I) was defined as:

$$I_{HxNy} = \mathbf{1}_{HxNy} * [f_{IHxNy} r_{IHxNy} + (1 - f_{IHxNy})] \qquad 3$$

## Likelihood

We used Eqs 1–3 to generate predicted case age distributions (p) for each influenza season (s) in which cases were observed in the data. Then, the likelihood was obtained as a product of multinomial densities across all seasons. If $n_s$ represents the total number of cases observed in a given season, $x_{0cs}, \ldots x_{mcs}$ each represent the number of cases observed in each single year of birth, and if $p_{0cs} \ldots p_{mcs}$ each represent entries in the model's predicted birth year-distribution of cases, then the likelihood is given by:

$$\mathcal{L} = \prod_s \frac{n_s!}{x_{0s}! \ldots x_{ms}!} p_{0s}^{x_{0s}} \cdots p_{ms}^{x_{ms}} \qquad 4$$

## Model fitting and model comparison

We fit models containing all possible combinations of the above factors to the surveillance data. We simultaneously estimated all free parameter values using the optim() function in R, with method L-BFGS-B. Imprinting parameters could take values in [0,1], representing the possibility of a reduction in risk. Age-specific risk parameters could take any value greater than 0. We calculated likelihood profiles and 95% profile confidence intervals for each free parameter. Confidence intervals were defined using the method of likelihood ratios [36].

## Antigenic advance

We obtained antigenic distance estimates from Nextstrain (nextstrain.org) [38,56], and from source data from Fig 3 in Bedford et al. [40]. Nextstrain calculates antigenic distance using genetic data from GISAID [57], and using methods described by Neher et al. [39]. We analyzed "CTiter" estimates from Nextstrain, which correspond to Neher et al.'s tree model method, and are most directly comparable to pre-2009 H1N1 estimates from [40]. We repeated analyses using estimates from Neher et al.'s substitution model method and verified that our choice of antigenic distance metric did not meaningfully impact our results. The negative Spearman correlation between antigenic advance and proportion of cases in children was lower, but still not statistically significant when using the substitution model (p = 0.06); all other differences were unremarkable. Datasets from Nextstrain and Bedford et al. both contained redundant antigenic distance estimates for the H3N2 subtype, but for subtype H1N1, only Bedford et al. analyzed the pre-2009 lineage, and only Nextstrain data analyzed the post-2009 lineage. The antigenic distance estimates reported by Bedford et al. were roughly proportional to those reported on Nextstrain, but greater in absolute magnitude [39]. To enable visualization of all lineages of H1N1 and H3N2 on the same plot axes, we rescaled pre-2009 H1N1 estimates from Bedford et al. using the formula $d_{Nextstrain} = 0.47 d_{Bedford}$. The scaling factor was chosen so that directly-comparable H3N2 distance estimates obtained using each method spanned the same range (S3 Fig). The Nextstrain data files used in this analysis are archived within our analysis code.

## Supporting information

**S1 Fig. ADHS age distributions, all seasons.** Supplement to Fig 2 showing observed age distributions from all influenza seasons. Observed case fractions (points) were only plotted if 10 or more cases of a given subtype were confirmed, to avoid extreme stretching of the y axis. Smoothing splines were only plotted if 50 or more cases of a given subtype were observed, as fits to fewer data points would not have been meaningful.
(TIFF)

**S2 Fig. Alternate smoothing parameters, AZDHS data.** Supplement to Fig 2, with smoothing parameters chosen to fit splines that are less (A-F), or more (G-L) smooth than the splines shown in the main text. Differences between H1N1 and H3N2's age-specific impacts remain evident, especially in the oldest cohorts, regardless of smoothness.
(PDF)

**S3 Fig. Comparison of rescaled antigenic distance estimates from the Bedford et al., and Nextstrain datasets.** Points represent average antigenic position of all isolates from a given calendar year.
(TIFF)

**S1 Data. PCR-confirmed cases from ADHS surveillance, and state-level census data.**
(ZIP)

## Acknowledgments

We are grateful to Ken Komatsu and Kristen Herrick for their assistance with data access, and to Trevor Bedford for assistance accessing and interpreting antigenic distance data from Nextstrain. We thank Lone Simonsen for helpful discussions.

## Code and data availability

Code to perform all reported analyses and construct all plots, and all relevant data (Arizona surveillance data and relevant antigenic advance data) is archived at https://zenodo.org/badge/latestdoi/160883450.

## Disclaimer

This work does not necessarily represent the views of the US government or the NIH.

## Author Contributions

**Conceptualization:** Katelyn M. Gostic, Cécile Viboud, Michael Worobey, James O. Lloyd-Smith.

**Data curation:** Rebecca Bridge, Shane Brady.

**Formal analysis:** Katelyn M. Gostic.

**Funding acquisition:** Katelyn M. Gostic, Michael Worobey, James O. Lloyd-Smith.

**Investigation:** Katelyn M. Gostic, Cécile Viboud, Michael Worobey, James O. Lloyd-Smith.

**Methodology:** Katelyn M. Gostic, Michael Worobey, James O. Lloyd-Smith.

**Project administration:** Katelyn M. Gostic, Rebecca Bridge, Shane Brady, James O. Lloyd-Smith.

**Resources:** Cécile Viboud, Michael Worobey, James O. Lloyd-Smith.

**Software:** Katelyn M. Gostic.

**Supervision:** Rebecca Bridge, Shane Brady, Cécile Viboud, Michael Worobey, James O. Lloyd-Smith.

**Validation:** Katelyn M. Gostic.

**Visualization:** Katelyn M. Gostic.

**Writing – original draft:** Katelyn M. Gostic.

**Writing – review & editing:** Rebecca Bridge, Shane Brady, Cécile Viboud, Michael Worobey, James O. Lloyd-Smith.

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
