## [Decision Letter · Decision Letter 0]

5 Aug 2019

Dear Ms. Gostic,

Thank you very much for submitting your manuscript "Childhood immune imprinting to influenza A shapes birth year-specific risk during seasonal H1N1 and H3N2 epidemics" (PPATHOGENS-D-19-01238) for review by PLOS Pathogens. Your manuscript was fully evaluated at the editorial level and by independent peer reviewers. The reviewers appreciated the attention to an important problem, but raised some substantial concerns about the manuscript as it currently stands. These issues must be addressed before we would be willing to consider a revised version of your study. We cannot, of course, promise publication at that time.

We therefore ask you to modify the manuscript according to the review recommendations before we can consider your manuscript for acceptance. Your revisions should address the specific points made by each reviewer.

(1) A letter containing a detailed list of your responses to the review comments and a description of the changes you have made in the manuscript. Please note while forming your response, if your article is accepted, you may have the opportunity to make the peer review history publicly available. The record will include editor decision letters (with reviews) and your responses to reviewer comments. If eligible, we will contact you to opt in or out.

(2) Two versions of the manuscript: one with either highlights or tracked changes denoting where the text has been changed; the other a clean version (uploaded as the manuscript file).

Additionally, to enhance the reproducibility of your results, PLOS recommends that you deposit your laboratory protocols in protocols.io, where a protocol can be assigned its own identifier (DOI) such that it can be cited independently in the future. For instructions see http://journals.plos.org/plospathogens/s/submission-guidelines#loc-materials-and-methods

We hope to receive your revised manuscript within 60 days. If you anticipate any delay in its return, we ask that you let us know the expected resubmission date by replying to this email. Revised manuscripts received beyond 60 days may require evaluation and peer review similar to that applied to newly submitted manuscripts.

[LINK]

Sincerely,

Sabra L. Klein

Guest Editor

PLOS Pathogens

Andrew Pekosz

Section Editor

PLOS Pathogens

Kasturi Haldar

Editor-in-Chief

PLOS Pathogens

orcid.org/0000-0001-5065-158X

Grant McFadden

Editor-in-Chief

PLOS Pathogens

orcid.org/0000-0002-2556-3526

All reviewers commented on the importance of addressing early life exposure in immune imprinting to influenza A viruses in humans. While the reviewers found the question important a few major concerns were raised that must be addressed. First, concerns were raised about biases in surveillance data collection, which can differ by age (Reviewer 3). Also, how host-related factors, including sex, in addition to age, could impact early life imprinting (Reviewer 1) is not considered in the the analyses to ensure that the conclusions drawn are valid across individuals.

Reviewer's Responses to Questions

**Part I - Summary**

Reviewer #1: The goal of this paper was to assess how childhood immune imprinting shapes seasonal influenza epidemiology. The authors use large epidemiological surveillance data to test whether this immune imprinting acts primarily by immune memory of a particular influenza subtype or via broader immunity that protects again various subtypes. The authors show that within subtype imprinting has the strongest impact on seasonal influenza risk and that antibody responses acquired later in life do not have the same impact on long-term immunity as early-life influenza responses. This is a well-written manuscript with important implications. My specific comments are below:

Reviewer #2: Overall this is a very insightful manuscript that describes the role of early life influenza exposures in protection against currently circulating seasonal influenza strains. This is a topic of great importance and the manuscript itself is well written with clearly presented data. The authors have published a previous paper examining how early life influenza exposures provide some degree of protection against potentially pandemic influenza strains. However, as humans are repeatedly exposed to seasonal influenza via both vaccination and infection, the protection afforded against circulating strains via imprinting could be vastly different. Indeed, the authors do find early life influenza exposures are able to imprint immunity and provide significant but only subtype-specific protection against future infections into adulthood and beyond. Further, a novel role for the NA protein in imprinting is described. Overall this paper represents an important contribution to the field and the data appear to support the conclusions being drawn.

Reviewer #3: This is a well-written manuscript by Gostic and colleagues examining imprinting, a critical question in the influenza field. The modeling is a real strength, but I have significant concerns about the data that is modeled.

**Part II – Major Issues: Key Experiments Required for Acceptance**

Reviewer #1: Assessment of host factors

Although the authors investigate how age can impact imprinting responses, they do not provide the sex of the individuals in their study, nor do they account for sex in their analyses. Previous work has shown that sex can impact influenza responses in human and animal models, and it is important that in this type of epidemiological study, host factors, including sex, are included. Further, although I recognize it is difficult to access human data over the life course, it would be important to know more about the patients included in the analysis and if any patients had co-morbidities that may have influenced their responses both early on and later in life. With increasing age, susceptibility to infection increases; antibody responses and vaccine efficacy decrease; and sex hormones decrease. These immunological and endocrinological changes are important to account for when assessing how age and year of birth may impact responses to influenza.

Reviewer #2: None

Reviewer #3: 1. This manuscript relies completely on influenza surveillance data, however, no detail is provided on the surveillance system. In particular, typically state surveillance systems are designed in such a way that they target specific at-risk populations. Thus, the data is collected in an intentionally biased way and does not represent what is happening at a population level, but rather maximizes the cases identified. For instance, sentinel surveillance systems tend to have over-representation of pediatric offices to catch non-severe cases, outbreaks tend to be investigated (and reported on) in old age homes, and hospital surveillance may be more heavily done in specific adult hospitals. In addition, the design of the surveillance system may change over time. If severity patterns differ by virus, which is likely, and surveillance setting differs by age (which is also likely, but I am unfamiliar with surveillance in Arizona), then bias could be introduced into the models. The authors should report on the design of the surveillance system in the methods section including any changes over the study period and, most importantly, on how the design of the surveillance system may have impacted their results.

**Part III – Minor Issues: Editorial and Data Presentation Modifications**

Reviewer #1: Childhood age range

The authors include “children” from 0 to 10 years of age. Have the authors thought about the fact that individuals within that large range may have varying degrees of imprinting and also exhibit vastly different immune responses (e.g., a 6-month old child will elicit different responses from a 10 year old). It would be helpful if the authors address this idea and suggest how they controlled for this in the current study.

Reviewer #2: While the discussion raises many important points, it is very long and would benefit from being compressed while still retaining the major points being made. Although outside of the scope of the current paper (and analyzed dataset), a future study performing similar analysis on a cohort on which serum/PBMCs could also be analyzed would provide significant insight on the immunologic mechanisms underlying the described findings.

Reviewer #3: 1. The authors should define imprinting more thoroughly in the introduction as non-influenza audiences may not be familiar with the term and in the influenza field imprinting has been used to describe several phenomena that may have different mechanistic bases (and indeed the authors themselves do this when using imprinting vs. within-subtype imprinting).

2. The authors state that HA group-level responses, which I assume includes the stem, are not thought to play a strong role in defense against familiar strains, however, a recent article by Ng et. al demonstrated that anti-HA stem antibodies are an independent correlate of protection.

3. Throughout the manuscript the authors refer to “infections” or “infecting” but they are looking at reported cases, not infections.

4. The authors suggest that imprinting may be a result of CD4 T-cells (as one possibility). Is there any evidence for this?

5. The authors use a case series. Did the population structure of Arizona change substantially over the time period? If so, they should adjust for the changes in the demographics.

PLOS authors have the option to publish the peer review history of their article (what does this mean?). If published, this will include your full peer review and any attached files.

Reviewer #1: No

Reviewer #2: No

Reviewer #3: No

---

## [Editor Report · Decision Letter 1]

25 Sep 2019

Dear Ms. Gostic,

We are pleased to inform that your manuscript, "Childhood immune imprinting to influenza A shapes birth year-specific risk during seasonal H1N1 and H3N2 epidemics", has been editorially accepted for publication at PLOS Pathogens. 

Before your manuscript can be formally accepted and sent to production, you will need to complete our formatting changes, which you will receive by email within a week. Please note that your manuscript will not be scheduled for publication until you have made the required changes.

IMPORTANT NOTES

(1) Please note, once your paper is accepted, an uncorrected proof of your manuscript will be published online ahead of the final version, unless you’ve already opted out via the online submission form. If, for any reason, you do not want an earlier version of your manuscript published online or are unsure if you have already indicated as such, please let the journal staff know immediately at plospathogens@plos.org.

(2) Copyediting and Proofreading: The corresponding author will receive a typeset proof for review, to ensure errors have not been introduced during production. Please review the PDF proof of your manuscript carefully, as this is the last chance to correct any errors. Please note that major changes, or those which affect the scientific understanding of the work, will likely cause delays to the publication date of your manuscript. 

(3) Appropriate Figure Files: Please remove all name and figure # text from your figure files. Please also take this time to check that your figures are of high resolution, which will improve the readbility of your figures and help expedite your manuscript's publication. Please note that figures must have been originally created at 300dpi or higher. Do not manually increase the resolution of your files. For instructions on how to properly obtain high quality images, please review our Figure Guidelines, with examples at: http://journals.plos.org/plospathogens/s/figures.

(4) Striking Image: Please upload a striking still image to accompany your article if one is available (you can include a new image or an existing one from within your manuscript). Should your paper be accepted, this image will be considered for our monthly issue image and may also appear on our website to feature your article. Please upload this as a separate file, selecting "striking image" as the file type upon upload. Please also include a separate "Other" file with a caption, including credits and any potential copyright information. Please do not include the caption in the main article file. If your image is from someone other than yourself, please ensure that the artist has read and agreed to the terms and conditions of the Creative Commons Attribution License at http://journals.plos.org/plospathogens/s/content-license. Please note that PLOS cannot publish copyrighted images.

(5) Press Release or Related Media: If your institution or institutions have a press office, please notify them about your upcoming paper at this point, to enable them to help maximize its impact. If they will be preparing press materials for this manuscript, please inform our press team in advance at plospathogens@plos.org as soon as possible. We ask that you contact us within one week to plan ahead of our fast Production schedule. If you need to know your paper's publication date for related media purposes, you must coordinate with our press team, and your manuscript will remain under a strict press embargo until the publication date and time. This means an early version of your manuscript will not be published ahead of your final version. 

(6)  PLOS requires an ORCID iD for all corresponding authors on papers submitted after December 6th, 2016. Please ensure that you have an ORCID iD and that it is validated in Editorial Manager.  To do this, go to ‘Update my Information’ (in the upper left-hand corner of the main menu), and click on the Fetch/Validate link next to the ORCID field.  This will take you to the ORCID site and allow you to create a new iD or authenticate a pre-existing iD in Editorial Manager

(7) Update your Profile Information: Now that your manuscript has been provisionally accepted, please log into Editorial Manager and update your profile, if needed. Go to https://www.editorialmanager.com/ppathogens, log in, and click on the "Update My Information" link at the top of the page. Please update your user information to ensure an efficient production and billing process. 

(8) LaTeX users only: Our staff will ask you to upload a TEX file in addition to the PDF before the paper can be sent to typesetting, so please carefully review our Latex Guidelines http://journals.plos.org/plospathogens/s/latex in the meantime.

(9) If you have associated protocols in protocols.io, please ensure that you make them public before publication to guarantee immediate access to the methodological details.

Best regards,

Sabra L. Klein

Guest Editor

PLOS Pathogens

Andrew Pekosz

Section Editor

PLOS Pathogens

Kasturi Haldar

Editor-in-Chief

PLOS Pathogens

orcid.org/0000-0001-5065-158X

Grant McFadden

Editor-in-Chief

PLOS Pathogens

orcid.org/0000-0002-2556-3526

The authors carefully considered the reviews and incorporated as many of the requested modifications as possible, given some limitations of the data set. This is a very important study that will make a significant impact on the field. There are no additional concerns or requests for modifications.
---

## [Editor Report · Acceptance letter]

8 Nov 2019

Dear Ms. Gostic,

We are delighted to inform you that your manuscript, "Childhood immune imprinting to influenza A shapes birth year-specific risk during seasonal H1N1 and H3N2 epidemics," has been formally accepted for publication in PLOS Pathogens.

Best regards,

Kasturi Haldar

Editor-in-Chief

PLOS Pathogens

orcid.org/0000-0001-5065-158X

Grant McFadden

Editor-in-Chief

PLOS Pathogens

orcid.org/0000-0002-2556-3526